# Oviposition Suitability of *Drosophila Suzukii* (Diptera: Drosophilidae) for Nectarine Varieties and Its Correlation with the Physiological Indexes

**DOI:** 10.3390/insects10080221

**Published:** 2019-07-24

**Authors:** Sha Liu, Huan-Huan Gao, Yi-Fan Zhai, Hao Chen, Hai-Yan Dang, Dong-Yun Qin, Li-Li Li, Qiang Li, Yi Yu

**Affiliations:** 1Institute of Plant Protection, Shandong Academy of Agricultural Sciences, Jinan 250100, China; 2College of Plant Protection, Yunnan Agricultural University, Kunming 650201, China; 3Shandong Academy of Grape, Jinan 250100, China; 4SAAS-CABI East Asia Joint Laboratory for Biocontrol, Jinan 250100, China; 5Shanxi Entry-Exit Inspection and Quarantine Bureau, Taiyuan 030000, China

**Keywords:** oviposition, firmness, soluble solids content, nutritional components

## Abstract

The nectarine is an important fruit, which is attacked by *Drosophila suzukii* in Europe and the United States but there are no reports of it attacking nectarines in China. Here, we determined the oviposition preference of *D. suzukii* six on intact and sliced nectarine varieties in China and how physical and physiological indexes of the fruit correlate with these preferences. *D. suzukii* were allowed to oviposit on two early–, two middle– and two late–maturing varieties of nectarine—Shuguang and Chunguang, Fengguang and Zhong you 4, Zhong you 7 and Zhong you 8, respectively and the number of larvae also followed the order. The firmness, soluble solids content and the nutritional components of the amino acid, protein, soluble sugar and pectin contents of each variety were measured. *D. suzukii* preferred the early Shuguang variety, followed by the early Chunguang variety and then the middle Zhong you 4 and Fengguang varieties. Taken together, results show that *D. suzukii* shows preferences for earlier rather than later varieties of nectarines in China and that these preferences are related to the fruit’s physical and physiological traits. Results suggest that mixed cultivation of early–, middle– and late–maturing nectarine varieties should be avoided in order to prevent fly dispersal and infestation by *D. suzukii*.

## 1. Introduction

*Drosophila suzukii* (Diptera: Drosophilidae) females have a tough serrated ovipositor, which can easily puncture the fresh intact fruit pericarp to oviposit in the pulp [1]. The larvae feeds on pulp, leading to infection of the fruit by bacteria and fungi, which then results in spoiled and rotten fruit [2]. One female can lay 400–600 eggs during its life. Depending on climate and weather conditions, 7 to 15 generations can be produced every year. With its fast generation time and ability to lay eggs on fresh fruit, *D. suzukii* causes a decline of yield and is a potential economic threat to its hosts including cherry, raspberry, blackberry, peach and nectarine [3,4]. It is widely distributed in North America, Europe and South America and it is expected to establish itself in Africa and Oceania, causing serious loss to the fruit industry, such as damage to several crops in North America, Europe [5,6] and Brazil [7], up to 80% economic loss for some varieties of grape and 100% for blueberry in Japan [8]. It is also found in multiple provinces in China [9]. There are over 60 host species of *D. suzukii*, including strawberry, cherry, blueberry, waxberry, grape and nectarine [10]. The damaging effects of *D. suzukii* on nectarines have been reported abroad. In Europe, *D. suzukii* harms nectarine growth, which has resulted in economic loss [11]. However, there are no reports on the damage of *D. suzukii* on nectarines in China yet. Nectarine is one of the most consumed fruits in Europe—oranges are the second most consumed fruit crop in Europe according to Food and Agriculture Organization—and it is one of the major fruits that are exported from China [12,13]. Therefore, it is important to understand the host-pest interactions of *D. suzukii* and nectarine in order to prevent damages to nectarine crops in China and worldwide.

Insect preference for oviposition substrate is closely related to the color, variety and maturity of the host fruit such as cherry and grape [14,15]. The oviposition behavior of *D. suzukii* depends on the variety and maturity of the fruit [16,17]. For example, females prefer to oviposit on varieties of cherries that have low firmness [18]. The nutritional and metabolic components of host plants affect the ovipositional preference of parasitoid and affect growth, development and reproduction of insects [19,20]. In particular, the soluble sugars, proteins and amino acids of host plants affect the ovipositional preference of parasitoid [21,22]. For instance, sugar and organic acid contents can affect fruit quality and the taste of the nectarine [23], *D. suzukii* prefers to oviposit on Yellow Honey and Red Light cherry varieties with relatively high amino acid content [18]. *D. simulans* prefers to oviposit on hosts with high protein content [24]. However, whether *D. suzukii* shows a preference for different nectarine varieties and fruit quality is currently unknown. Therefore, experiments on the ovipositional preference of *D. suzukii* on intact and sliced nectarines were carried out.

Nectarines are cultivated under open conditions, resulting in variable quality indexes, such as color, maturity, rigidity and soluble solids, between different nectarine varieties at the same period [25,26,27,28]. To understand how variability in the quality of fruit can affect the oviposition patterns of *D. suzukii*, we investigated physiological indexes among different nectarine varieties under open cultivation conditions at the same period and measured the ovipositional preference and egg-laying amount of *D. suzukii* on sliced nectarine varieties over time as the fruit matured. We tested whether there were correlations between *D. suzukii* oviposition patterns and nectarine variety, maturity, firmness, soluble solid, amino acid, protein, soluble sugar and pectin contents. Our study provides a foundation for selecting nectarine varieties to prevent damage caused by *D. suzukii*.

## 2. Material and Methods

### 2.1. Insect and Nectarine

Six nectarine varieties were examined in this study, including Shuguang and Chunguang (early–maturing variety), Zhong you 4 and Fengguang (medium-maturing variety), Zhong you 7 and Zhong you 8 (late–maturing variety), fruits were collected from Xincao Village, Peach village Town, Salt Lake District, in Yuncheng City, Shanxi Province (111°10′ 35″ E, 35°17′ 15″ N). Shuguang and Chunguang fruits were picked on June 10th and June 16th and Zhong you 4 was picked on June 10th, June 16th, June 23rd, June 30th and July 7th. Other varieties were picked every seven days from June 10th to August 4th, nine times in total. All varieties were picked in 2018. *D. suzukii* adults were obtained from the laboratory of the Institute of Plant Protection, Shandong Academy of Agricultural Sciences and the colony was reared in a controlled room at 25 ± 1 °C, 70 ± 5% RH and 16: 8 h (L: D) [29] with an artificial diet composed of corn flour, mashed apple and banana, yeast, sucrose, agar and sorbitol described by Zhai et al. [30].

### 2.2. Oviposition Preference of D. suzukii on Six Nectarine Varieties

Oviposition preference test experiments on *D. suzukii* were conducted indoors by introducing flies to intact nectarines. With each variety, the non-hurt nectarines with the same ripeness and 40 *D. suzukii* females that mated (for 3 days) were placed into a tissue-culture bottle. After 24 hours, females were dispersed, eggs laid on each variety were respectively counted using light microscopy Olympus CX41RF (Olympus Corporation, TOKYO). Then, the bottles were placed in a climate-controlled growth chamber at 25 ± 0.5 °C with 70 ± 0.5% relative humidity (RH) and a photoperiod of 16:8 h (L: D), the number of larvae in the nectarine samples was recorded. Four replicates were tested for each variety.

For the sliced oviposition preference experiment of *D. suzukii*, different non-hurt varieties of nectarine with the same size and ripeness were selected and cut into 20g slices near the peel and placed into the same glass tissue-culture bottle mentioned above. Other procedures followed those of the experiments of oviposition non-choice on intact nectarines.

### 2.3. Physiological Indexes of Different Nectarine Varieties

#### 2.3.1. Firmness

A fruit schlerometer (GY-1, Zhejiang Top Instrument Co. Ltd., Zhejiang, China) was used to measure the firmness of nectarines. Measurements were taken by inserting the cylindrical probe into the fruit for about 10 mm and 10 nectarines were used to examine the firmness of each variety per date. Ten replicates were tested for each variety.

#### 2.3.2. Soluble Solid Content

A temperature-compensated refractometer (Texas runxin Instrument Co. Ltd., China) was used to measure the soluble solids content of the nectarines at 20 °C. Different varieties of nectarine with the same size and ripeness were selected and cut into 20 g slices near the pericarp. A juice extractor (JYL-C051, Beijing jiuyang electric appliance co. LTD., Beijing, China) was used to grind the slices and the juice of slices from ten nectarines was centrifuged at 8000 rpm for 10 min at room temperature. Two drops of the liquid supernatant was placed on a refractometer and the scale of the refractometer was recorded as the soluble solids content. Ten replicates were tested for each variety per date.

#### 2.3.3. Amino Acid, Protein, Soluble Sugar and Pectin Content

Approximately 0.3 g from near the pericarp of the fruit from five samples per variety and date were collected. The amino acid, protein, soluble sugar and pectin content were determined by the process of lapping, centrifuging, adding reagent, water bath. Ultimately, the colorimetric value was determined with an EMax Plus Microplate Reader under 570 nm, 562 nm, 620 nm and 530 nm wavelength respectively according to the following kits: amino acid: AA-1-W; protein: BCAP-1-W; soluble sugar; pectin: WSP-1-Y (Suzhou Comin biotechnology Co., Ltd., Suzhou, China).

### 2.4. Statistical Analyses

The results of the ovipositional preference of *D. suzukii* to the nectarine varieties, the firmness, the soluble solid, amino acid, protein, soluble sugar and pectin contents were respectively analyzed with one-way analysis of variance, followed by Student-Newman-Keuls test using the SolutionsStatistical Package for the Social Sciences 19.0 software. Comparisons were only made for the nectarine fruits that were collected on the same day. Relationship between oviposition preference and physiological indexes was analyzed by a *Pearson* correlation analysis with 0.05 as the cut-off for significance, *Pearson* correlation below was replaced by “*r*”.

## 3. Results

### 3.1. Oviposition Preference of D. suzukii on Intact Nectarine

We conducted the oviposition preference trial where females were allowed to oviposit on intact fruit collected on the same date (Table 1). On June 10th and 16th, *D. suzukii* preferred to lay eggs on Shuguang than Chunguang nectarines (June 10th: *F*_5,12_ = 66.85, *p* < 0.01; June 16th: *F*_5,12_ = 53.18, *p* < 0.01). On June 23rd and 30th, *D. suzukii* did not oviposit on any variety. On July 7th, *D. suzukii* laid more eggs on Zhong you 4 than Fengguang nectarines and preferred to lay significantly more eggs on Zhong you 4 (July 7th: *F*_3,8_ = 61.70, *p* < 0.01). When Zhong you 4 nectarines were completely harvested on July 13th, *D. suzukii* chose to oviposit on Fengguang nectarines (*F*_2,6_ = 83.31, *p* < 0.01). On July 20th, *D. suzukii* began to oviposit on Zhong you 7 intact nectarines but it laid more eggs on Fengguang nectarines than Zhong you 7 (*F*_2,6_ = 19.88, *p* < 0.01). As the Zhong you 8 matured on July 27th, *D. suzukii* began to oviposit on it but females still laid more eggs on Zhong you 7 intact fruit than Fengguang (*F*_2,6_ = 67.79, *p* < 0.01). On August 4th, there were no significant differences in the amount of eggs laid by Fengguang and Zhong you 7 intact nectarines, the amount of eggs on Zhongyou 8 was significantly less than the other varieties (*F*_2,6_ = 91.65, *p* < 0.01).

The number of larvae on different intact nectarine varieties is shown in Table 2. The results performed the similar regular compared with Table 1, with the ripeness of the nectarine, the number of larvae was increasing. Between June 10th and June 16th, only Shuguang and Chunguang nectarines were chosen to lay eggs and the number of larvae on Shuguang nectarines was more than on Chunguang. However, there were no larvae found on each nectarine variety on June 23th and June 30th, Shuguang and Chunguang nectarine had been harvested. The number of larvae laid on Zhong you 4 was more than on Fengguang on July 7th. There were more larvae on the Fengguang nectarine on July 13th and July 20th and the number of larvae on Zhong you 7 was more than on any other variety. On July 27th and August 4th, *D. suzukii* showed more preference for Zhong you 7 and the amount of larvae on Zhongyou 8 was less than the other varieties.

### 3.2. Oviposition Preference of D. suzukii on Sliced Nectarine

The trial on the oviposition preference of *D. suzukii* was conducted using sliced fruit. We found that *D. suzukii* had significant preferences for certain sliced nectarine varieties (Table 3). On June 10th and 16th, females laid more eggs on Shuguang sliced nectarines, followed by Chunguang and Zhong you 4 (June 10th: *F*_5,12_ = 170.80, *p* < 0.01; June 16th: *F*_5,12_ = 257.15, *p* < 0.01). On June 23rd, *D. suzukii* laid significantly more eggs on Fengguang sliced nectarines than on Zhong you 4 (*F*_3,8_ = 38.62, *p* < 0.01). On June 30th and July 7th, the sliced Zhong you 4 had been the preferred variety for *D. suzukii* to lay eggs over the other three varieties. On July 7th, *D. suzukii* began to oviposit on Zhong you 7 sliced nectarines but females laid significantly fewer eggs on Zhong you 7 compared to Fengguang and Zhong you 4 (June 30th: *F*_3,8_ = 96.80, *p* < 0.01; July 7th: *F*_3,8_ = 201.89, *p* < 0.01). When Zhong you 4 nectarines were harvested on July 13th and 20th, the females had laid more eggs on sliced Fengguang nectarines, followed by Zhong you 7 and Zhong you 8 (July 13th: *F*_2,6_ = 41.66, *p* < 0.01; July 20th: *F*_2,6_ = 37.50, *p* < 0.01). On July 27th and August 4th, the egg-laying amount of *D. suzukii* was similar for Zhong you 7 and Fengguang nectarines but both of them were significantly higher than for Zhong you 8 sliced nectarines (July 27th: *F*_2,6_ = 50.01, *p* < 0.01; August 4th: *F*_2,6_ = 17.82, *p* < 0.01).

The number of larvae on different sliced nectarine varieties is shown in Table 4. The number of larvae increased with the ripeness of the nectarine. Between June 10th and June 16th, the number of larvae on Shuguang was the most of any other variety. On June 23th, there were more larvae on Zhongyou 4 on June 30th and July 7th. The number of larvae on the Fengguang nectarine was more than on the other varieties—the amount of larvae on Zhongyou 8 was lowest. On August 4th, there were more larvae on Zhongyou 7.

### 3.3. The Fruit Firmness and Soluble Solid Contents of Nectarine

From Table 5, we could easily observe that, as the variety became mature, the firmness declined gradually. On June 10th and 16th, the firmnesses of Shuguang and Chunguang nectarines were significantly lower than that of the other varieties (June 10th: *F*_5,12_ = 37.79, *p* < 0.01; June 16th: *F*_5,12_ = 49.92, *p* < 0.01). The firmness of Fengguang, Zhong you 7 and Zhong you 8 was similar on June 23rd but all were significantly more rigid than Zhong you 4 (*F*_3,8_ = 44.49, *p* < 0.01). On June 30th, July 7th and July 13th, there were significant differences for the firmness among all the varieties (June 30th: *F*_3,8_ = 59.19, *p* < 0.01; July 7th: *F*_3,8_ = 95.08, *p* < 0.01; July 13th: *F*_2,6_ = 82.74, *p* < 0.01). On July 20th and 27th, no significant difference was found in firmness between Fengguang and Zhong you 7 (July 20th: *F*_2,6_ = 242.46, *p* < 0.01; July 27th: *F*_2,6_ = 122.66, *p* < 0.01) and on August 4th, the firmness of Fengguang nectarines was significantly lower than the other varieties (*F*_2,6_ = 104.19, *p* < 0.01). Zhong you 8 was significantly more rigid than other varieties during the experimental stages. We conducted a Pearson correlation analysis on the oviposition of *D. suzukii* with the firmness of the fruit and we found significant negative correlations between firmness and oviposition (oviposition preference on intact nectarine: *r* = −0.83, *p* < 0.01, sliced nectarine: *r* = −0.90, *p* < 0.01, Table 8).

As the variety became mature, the soluble solid contents increased gradually (Table 4). On June 10th, the soluble solid content of Chunguang was significantly higher than other varieties (*F*_5,12_ = 7.95, *p* < 0.01). On June 16th, no significant difference was found in the soluble solid contents among the varieties (*F*_5,12_ = 2.97, *p* = 0.06). However, the soluble solid content of Fengguang nectarines was significantly lower than other varieties on June 23rd and 30th (June 23rd: *F*_3,8_ = 4.84, *p* = 0.03; June 30th: *F*_3,8_ = 9.80, *p* = 0.01). On July 7th, Fengguang nectarines had a lower soluble solid content compared to Zhong you 4 and Zhong you 8 (*F*_3,8_ = 31.07, *p* < 0.01). On July 13th and 20th, Zhong you 7 had a significantly higher soluble solid content than Fengguang and Zhong you 8 (July 13th: *F*_2,6_ = 9.88, *p* = 0.01; July 20th: *F*_2,6_ = 7.23, *p* = 0.03). However, there was no obvious difference among different varieties on July 27th and August 4th (July 27th: *F*_2,6_ = 4.19, *p* = 0.07; August 4th: *F*_2,6_ = 3.16, *p* = 0.12). The results of Pearson correlation analysis revealed that there was a significant positive correlation between the oviposition preference of *D. suzukii* and the soluble solid content of nectarines (oviposition preference on intact nectarine: *r* = 1.00, *p* = 0.01, sliced nectarine: *r* = 1.00, *p* = 0.002, Table 8). We also found a significant negative correlation between firmness and soluble solid contents of nectarine (*r* = −0.278, *p* < 0.01).

### 3.4. Physiological Indexes of Nectarines

#### 3.4.1. The Amino Acid and the Protein Content

Nectarines that were collected on the same day had significant differences in amino acid contents depending on the nectarine variety (Table 6). In general, the amino acid contents of Shuguang, Chunguang, Zhong you 4 and Fengguang were higher than other varieties when collected on the same day. On June 10th and 16th, the amino acid contents of Shuguang and Chunguang nectarines were significantly higher than any other variety (June 10th: *F*_5,12_ = 602.21, *p* < 0.01; June 16th: *F*_5,12_ = 1560.62, *p* < 0.01). The amino acid contents of Zhong you 4 nectarines had been higher than other varieties significantly from June 23rd to July 7th, which was at least six times more than Fengguang, Zhong you 7 and Zhong you 8 nectarines (June 23rd: *F*_3,8_ = 2011.50, *p* < 0.01; June 30th: *F*_3,8_ = 1240.67, *p* < 0.01; July 7th: *F*_3,8_ = 2710.96, *p* < 0.01).

From July 13th to August 4th, the amino acid content in Fengguang nectarine was significantly higher than Zhong you 7 and Zhong you 8 (July 13th: *F*_2,6_ = 136.10, *p* < 0.01; July 20th: *F*_2,6_ = 225.86, *p* < 0.01; July 27th: *F*_2,6_ = 380.60, *p* < 0.01; August 4th: *F*_2,6_ = 161.10, *p* < 0.01). Through Pearson correlation analyses, we found that the amount of eggs laid on *D. suzukii* on nectarine fruits had a significantly positive correlation with amino acid content (oviposition preference on intact nectarine: *r* = 0.57, *p* < 0.01, sliced nectarine: *r* = 0.70, *p* < 0.01, Table 8). There was also a significant negative correlation between firmness and amino acid content of nectarine (*r*= −0.76, *p* < 0.01), as well as a significant positive correlation between amino acid and soluble solids content (*r* = 0.22, *p* = 0.02).

We observed significant differences in the protein content of nectarines depending on the variety and the collecting time. On June 10th, there was no significant difference in protein content among Shuguang, Chunguang, Zhong you 4 and Fengguang nectarines but which was significantly higher than Zhong you 7 and Zhong you 8 (*F*_5,12_ = 16.94, *p* < 0.01). The protein content of Zhong you 8 on June 16th was significantly higher than other varieties (*F*_5,12_ = 7.51, *p* < 0.01). However, the protein content in Zhong you 4 was the highest among those varieties collected on June 23rd, reaching up to 88.83 ± 5.95 mg/g (*F*_3,8_ = 43.12, *p* < 0.01). On June 30th, the protein contents of Fengguang and Zhong you 8 nectarines were relatively higher than other varieties (*F*_3,8_ = 6.65, *p* = 0.01). Then the protein contents in each nectarine variety gradually increased after June 30th. The protein content of Fengguang nectarines was significantly lower than Zhong you 7 and Zhong you 8 nectarines on July 13th and 20th (July 13th: *F*_2,6_ = 21.18, *p* < 0.01; July 20th: *F*_2,6_ = 6.09, *p* = 0.04) but obviously increased on July 27th and August 4th, (July 27th: *F*_2,6_ = 24.68, *p* < 0.01; August 4th: *F*_2,6_ = 66.75, *p* < 0.01). In general, after June 10th, the protein content of Zhong you 8 and Fengguang nectarines was higher than the other varieties. According to the Pearson correlation analysis of protein content with other parameters that were measured, we found a significant positive correlation between protein content and amino acid content (*r* = 0.21, *p* = 0.028) and soluble solids content (*r* = 0.73, *p* < 0.01) but a significant negative correlation with firmness (*r* = −0.21, *p* = 0.026). There was a significantly positive correlation between protein content and oviposition preference (On intact nectarines: *r* = 0.30, *p* = 0.08; sliced nectarines: *r* = 0.42, *p* = 0.01, Table 8).

#### 3.4.2. The Soluble Sugar and the Pectin Content

The significant differences in soluble sugar contents among different nectarine varieties are shown in Table 7. Overall, the soluble sugar contents of Shuguang, Chunguang and Zhong you 7 were higher than other varieties. On June 10th, the soluble sugar content of Shuguang nectarine was significantly higher than that of other varieties (*F*_5,12_ = 548.41, *p* < 0.01). On June 16th, the soluble sugar content of Chunguang nectarines was the highest, reaching 64.38±1.20 mg/g, followed by Shuguang, Zhong you 4 and Fengguang (*F*_5,12_ = 1640.89, *p* < 0.01). The soluble sugar content of Zhong you 4 nectarines was 51.44 ± 2.23 mg/g on July 7th, which was significantly higher than other varieties (June 23rd: *F*_3,8_ = 294.03, *p* < 0.01; June 30th: *F*_3,8_ = 109.52, *p* < 0.01; July 7th: *F*_3,8_ = 170.01, *p* < 0.01). Between July 13th and August 4th, the soluble sugar content increased for each variety and the soluble sugar contents of Fengguang and Zhong you 4 nectarines were significantly higher than Zhong you 8 (July 13th: *F*_2,6_ = 7.40, *p* = 0.024; July 20th: *F*_2,6_ = 29.99, *p* < 0.01; July 27th: *F*_2,6_ = 125.90, *p* < 0.01; August 4th: *F*_2,6_ = 40.66, *p* < 0.01). The results of Pearson correlation analysis showed that a significant positive correlation between the oviposition preference of *D. suzukii* and soluble sugar content (oviposition preference on intact nectarines: *r* = 0.74, *p* < 0.01, on sliced nectarines: *r* = 0.80, *p* < 0.01, Table 8). The soluble sugar contents in nectarines had a significant negative correlation with firmness (*r* = −0.878, *p* < 0.01) and a significant positive correlation with soluble solid content (*r* = 0.287, *p* < 0.01) and amino acid content of the nectarine (*r* = 0.835, *p* < 0.01).

The significant differences in pectin contents among different nectarine varieties collected on the same day are shown in Table 7. On June 10th, the pectin contents of Zhong you 7, Chunguang and Zhong you 4 nectarines were significantly higher than Shuguang, Fengguang and Zhong you 8 nectarines (*F*_5,12_ = 6.83, *p* < 0.01). There was no significant difference for the pectin contents among Shuguang, Chunguang and Zhong you 4 nectarines on June 16th but they were significantly higher than Fengguang, Zhong you 7 and Zhong you 8 (*F*_5,12_ = 51.35, *p* < 0.01). The pectin contents of Zhong you 7 and Zhong you 8 increased on June 30th, which were significantly higher than Fengguang and Zhong you 4 (*F*_3,8_ = 73.24, *p* <0.01). On July 7th, the pectin content of each variety decreased slightly and the pectin content of Zhong you 8 was significantly higher than that of Zhong you 4 and Zhong you 7 (*F*_3,8_ = 7.25, *p* = 0.01). Until July 20th, the pectin content of Zhong you 7 was lower than Fengguang nectarines (July 13th: *F*_2,6_ = 6.91, *p* = 0.03; July 20th: *F*_2,6_ = 7.72, *p* = 0.02). On August 4th, the pectin contents in Fengguang and Zhong you 7 were significantly higher than Zhong you 8 (*F*_2,6_ = 21.79, *p* < 0.01). Pearson correlation analyses revealed that the eggs of *D. suzukii* on nectarines were not correlated with pectin content (eggs on intact nectarines: *r* = 0.02, *P* = 0.89, sliced nectarines: *r* = −0.12, *P* = 0.48, Table 8). The pectin content of nectarine showed a significant negative correlation with protein content (*r* = −0.29, *p* < 0.01) and soluble solid content (*r* = −0.23, *P* = 0.02) and was not correlated with fruit firmness (*r* = 0.01, *p* = 0.93).

## 4. Discussion

In this study, we found that there were significant differences in the oviposition preference of *D. suzukii* on different nectarine varieties that were collected on the same day in an open field. *D. suzukii* females showed oviposition preference on sliced Shuguang nectarines, followed by Chunguang, Fengguang, Zhong you 4 and Zhong you 7 nectarine. Taken together, under the current cultivation pattern and ripening period, during the trial, *D. suzukii* mostly preferred to oviposit on Shuguang, Chunguang, Fengguang and Zhong you 4 nectarines.

Based on the fruits that were preferred by *D. suzukii*, we suspect that olfactory cues may have played an important role in locating oviposition substrates [15]. The varieties of nectarines that matured showed a significant effect on the choice of *D. suzukii*. Previous studies had shown that *D. suzukii* showed a preference for different grape varieties [31], with the mature grape varieties being more attractive for *D. suzukii* to oviposit in [32]. Similarly, intact guavas were much easier for *D. suzukii* to oviposit in [33]. Therefore, our results were similar to previous work on *D. suzukii* oviposition preferences. When measured on the same day, *D. suzukii* preferred to oviposit on mature nectarines, with no larvae found on the immature nectarine varieties, suggesting that fruit maturity was also an important factor for *D. suzukii* females when choosing oviposition sites.

As fruits matured, they became softer [34,35]. The ovipositional preference of insects is closely correlated with fruit firmness. For example, *Bactrocera dorsalis* and *Bactrocera cucurbitae* preferred to oviposit on mature and softer host fruits. *D. suzukii* preferred to oviposit on blueberry varieties and soft-peel tomatoes which were softer [36]. Here, we found that the proportion of chosen oviposition and the amount of oviposition of *D. suzukii* on different nectarine varieties were all negatively correlated with fruit firmness. On June 10th, *D. suzukii* only oviposited on Shuguang, Chunguang and Zhong you 4 nectarines that were softer. Between June 16th to July 7th, Shuguang and Chunguang nectarines were completely harvested and Fengguang and Zhong you 4 nectarines were becoming more mature. During this period, *D. suzukii* began to choose to oviposit on Zhong you 4 intact nectarines that were softer. The firmness of Fengguang, Zhong you 4 and Zhong you 7 intact nectarines were lower than 8.52 ± 0.39 kg/cm^2^, 4.54 ± 0.20 kg/cm^2^ and 6.87 ± 0.34 kg/cm^2^, respectively and *D. suzukii* began to damage the fruits through its oviposition. There are many nectarine varieties cultivated in China and we focused on the most popular varieties that are cultivated widely. However, the oviposition preference of *D. suzukii* for other varieties of nectarines still requires further study.

Aside from firmness, the nutritional content of the fruit also changed as the fruit became mature [24]. Nutrients contained in the fruit likely influenced the growth and development of insect larvae [20]. To sustain the insect population and the development of its offspring, insects select hosts with high nutrient levels [24,37]. Gao et al. [18] found that the nutritional conditions affected the oviposition preference of *D. suzukii* to different cherry varieties. We suggest that *D. suzukii* females preferred to oviposit on nectarine varieties with high soluble sugar and amino acids. Amino acids provide nitrogen and soluble sugar provided carbohydrates for larvae development [38,39]. Host plants with high amino acid and soluble sugar contents were less resistant to *D. suzukii*. Similar results have been found in cucumber varieties [40] and that varieties with high total sugar and amino acid were more likely to be infested by *B. cucurbitae*.

Therefore, during the cultivation process and ripening period of nectarines, the late–maturing varieties, such as Zhong you 7 and Zhong you 8, show higher resistance to *D. suzukii* than other varieties. Similar results were found with *Anastrepha fraterculus* in peaches in Brazil [41]. This suggests that early–maturing varieties (Shuguang and Chunguang) and medium–maturing varieties (Zhong you 4 and Fengguang) should be cultivated early, so that the orchard can avoid peak *D. suzukii* population levels and improve yield and quality of nectarines. The mixed cultivation of early–maturing, medium–maturing and late–maturing varieties should be avoided in order to prevent dispersal and infestations of *D. suzukii*.

## 5. Conclusions

In conclusion, under the open cultivation pattern, *D. suzukii* oviposited first on Shuguang, followed by Chunguang, Zhong you 4, Fengguang, Zhong you 7 and Zhongyou 8; what is more, the number of larvae on different nectarine varieties also corresponded with it, which increases with the ripeness of the nectarine and varies as the physiological indexes changes. It is reasonable for early–maturing varieties (Shuguang and Chunguang) and medium–maturing varieties (Zhong you 4 and Fengguang) to plant earlier in order that they are not susceptible of *D. suzukii*, moreover, it should be avoided for mixed cultivation. 

## Figures and Tables

**Table 1 insects-10-00221-t001:** The amount of eggs of D. suzukii on intact nectarine varieties on the same collection date.

Date	Number of Eggs/Head					
	Shuguang	Chunguang	Fengguang	Zhongyou 4	Zhongyou 7	Zhongyou 8
6.10	13.50 ± 1.19a	6.50 ± 1.19b	0c	0c	0c	0c
6.16	16.75 ± 1.93a	10.00 ± 1.47b	0c	0c	0c	0c
6.23	-	-	0	0	0	0
6.30	-	-	0	0	0	0
7.7	-	-	4.00 ± 0.91b	13.75 ± 1.38a	0c	0c
7.13	-	-	9.50 ± 1.04a	-	0b	0b
7.20	-	-	9.75 ± 1.55a	-	5.75 ± 1.10b	0c
7.27	-	-	12.25 ± 1.11b	-	18.00 ± 1.08a	2.75 ± 0.48c
8.4	-	-	21.25 ± 1.38a	-	22.75 ± 1.25a	3.25 ± 0.63b

Different letters next to the values (mean ± SE) in the same row indicate significant differences for oviposition preference among different intact varieties on the same collection date that was examined using the S-N-K test (*p* < 0.05). Negative sign indicates the variety was harvested entirely and there were no data points for those dates, the same as following.

**Table 2 insects-10-00221-t002:** The amount of larvae of *D. suzukii* on intact nectarine varieties on the same collection date.

Date	Number of Larvae/Head					
	Shuguang	Chunguang	Fengguang	Zhongyou 4	Zhongyou 7	Zhongyou 8
6.10	11.00 ± 0.41a	5.00 ± 1.22b	0c	0c	0c	0c
6.16	14.75 ± 2.59a	9.00 ± 0.91b	0c	0c	0c	0c
6.23	-	-	0	0	0	0
6.30	-	-	0	0	0	0
7.7	-	-	3.75 ± 0.75b	12.25 ± 1.11a	0c	0c
7.13	-	-	8.75 ± 0.63a	-	0b	0b
7.20	-	-	8.50 ± 1.32a	-	5.50 ± 0.96a	0b
7.27	-	-	11.50 ± 1.19b	-	16.75 ± 0.95a	2.75 ± 0.48c
8.4	-	-	18.75 ± 1.80a	-	21.25 ± 1.25a	3.00 ± 0.71b

**Table 3 insects-10-00221-t003:** The amount of eggs of *D. suzukii* on sliced nectarine varieties on the same collection date.

Date	Number of Eggs/Head					
	Shuguang	Chunguang	Fengguang	Zhongyou 4	Zhongyou 7	Zhongyou 8
6.10	25.25 ± 1.25a	17.50 ± 1.50b	0c	1.25 ± 0.75c	0c	0c
6.16	29.50 ± 1.32a	27.50 ± 1.04a	2.25 ± 0.75b	3.25 ± 1.11b	0b	0b
6.23	-	-	8.25 ± 1.11a	4.25 ± 0.63b	0c	0c
6.30	-	-	13.75 ± 1.11b	19.25 ± 1.65a	0c	0c
7.7	-	-	16.25 ± 0.95b	33.25 ± 1.70a	3.25 ± 0.85c	0c
7.13	-	-	22.50 ± 1.85a	-	17.00 ± 0.91b	6.00 ± 0.91c
7.20	-	-	29.75 ± 1.80a	-	20.75 ± 1.25b	12.00 ± 1.22c
7.27	-	-	34.50 ± 1.55a	-	31.75 ± 1.44a	17.00 ± 0.91b
8.4	-	-	35.75 ± 1.80a	-	35.50 ± 1.94a	22.25 ± 1.75b

Different letters next to the values (mean ± SE) in the same row indicate significant differences for eggs laid by *D. suzukii* among different sliced varieties on the same collection date that was examined using the S-N-K test (*p* < 0.05). Negative sign indicates the variety was harvested entirely and there were no data points for those dates.

**Table 4 insects-10-00221-t004:** The amount of larvae of *D. suzukii* on sliced nectarine varieties on the same collection date.

Date	Number of larvae/head					
	Shuguang	Chunguang	Fengguang	Zhongyou 4	Zhongyou 7	Zhongyou 8
6.10	22.25 ± 0.85a	16.25 ± 1.65b	0c	1.00 ± 0.58c	0c	0c
6.16	25.50 ± 1.04a	25.25 ± 1.55a	1.50 ± 0.29b	2.50 ± 0.87b	0b	0b
6.23	-	-	6.25 ± 0.48a	2.75 ± 0.75b	0c	0c
6.30	-	-	11.00 ± 1.08b	17.75 ± 1.65a	0c	0c
7.7	-	-	13.75 ± 1.11b	30.50 ± 1.71a	2.75 ± 0.63c	0c
7.13	-	-	21.00 ± 1.96a	-	16.00 ± 0.91b	5.25 ± 1.03c
7.20	-	-	26.75 ± 2.32a	-	18.75 ± 1.18b	10.75 ± 1.38c
7.27	-	-	32.50 ± 1.04a	-	30.75 ± 1.70a	16.00 ± 1.29b
8.4	-	-	30.75 ± 2.25a	-	34.00 ± 2.20a	20.25 ± 1.93b

Different letters next to the values (mean ± SE) in the same row indicate significant differences for larvae among different sliced varieties on the same collection date that was examined using the S-N-K test (*p* < 0.05). Negative sign indicates the variety was harvested entirely and there were no data points for those dates.

**Table 5 insects-10-00221-t005:** The firmness and soluble solids content of nectarine varieties.

Index	Date	Shuguang	Chunguang	Fengguang	Zhongyou 4	Zhongyou 7	Zhongyou 8
Firmness (kg/cm^2^)	6.10	4.76 ± 1.17b	4.47 ± 1.41b	14.2 ± 0.13a	13.99 ± 0.31a	14.98 ± 0.83a	15.03 ± 0.19a
	6.16	4.46 ± 1.16c	3.88 ± 1.04c	13.84 ± 0.57a	10.47 ± 0.36b	14.55 ± 0.17a	14.94 ± 0.40a
	6.23	-	-	13.43 ± 0.71a	9.04 ± 0.07b	13.85 ± 0.20a	14.68 ± 0.16a
	6.30	-	-	9.33 ± 0.04c	7.76 ± 0.32d	12.43 ± 0.53b	14.53 ± 0.49a
	7.7	-	-	8.52 ± 0.39c	4.54 ± 0.20d	11.26 ± 0.60b	14.24 ± 0.41a
	7.13	-	-	7.74 ± 0.47c	-	10.47 ± 0.25b	13.78 ± 0.23a
	7.20	-	-	7.39 ± 0.15b	-	6.87 ± 0.34b	13.12 ± 0.10a
	7.27	-	-	4.79 ± 0.34b	-	6.03 ± 0.55b	12.64 ± 0.12a
	8.4	-	-	3.52 ± 0.26c	-	5.37 ± 0.15b	11.53 ± 0.65a
Soluble solid content (%)	6.10	8.82 ± 0.16b	10.28 ± 0.15a	8.53 ± 0.26b	9.13 ± 0.26b	9.16 ± 0.30b	8.58 ± 0.19b
	6.16	11.17 ± 0.62a	11.52 ± 0.48a	10.08 ± 0.33a	10.81 ± 0.15a	11.68 ± 0.23a	10.32 ± 0.20a
	6.23	-	-	10.28 ± 0.65b	11.25 ± 0.41ab	12.30 ± 0.23a	11.96 ± 0.14a
	6.30	-	-	11.02 ± 0.37b	12.22 ± 0.48a	13.48 ± 0.23a	12.87 ± 0.20a
	7.7	-	-	11.14 ± 0.24c	13.28 ± 0.45b	14.79 ± 0.17a	13.64 ± 0.10b
	7.13	-	-	12.78 ± 0.39b	-	15.25 ± 0.53a	13.75 ± 0.20b
	7.20	-	-	13.36 ± 0.20b	-	15.44 ± 0.59a	14.23 ± 0.26ab
	7.27	-	-	13.92 ± 0.57a	-	15.86 ± 0.54a	14.82 ± 0.25a
	8.4	-	-	14.67 ± 0.78a	-	16.87 ± 0.69a	15.30 ± 0.36a

Different letters next to the values (mean ± SE) in the same row indicate significant differences for firmness and soluble solid content among different varieties on the same collection date that was examined using the S-N-K test (*p* < 0.05). Negative sign indicates the variety was harvested entirely and there were no data points for those dates.

**Table 6 insects-10-00221-t006:** The amino acids and protein content of nectarine varieties.

Index	Date	Shuguang	Chunguang	Fengguang	Zhongyou 4	Zhongyou 7	Zhongyou 8
Amino acids content (μmol/g)	6.10	75.47 ± 1.34 a	74.03 ± 1.40 a	4.70 ± 0.47 d	58.39 ± 2.53 b	11.89 ± 0.66 c	9.32 ± 0.78 c
	6.16	100.95 ± 2.03 a	104.38 ± 1.47 a	4.37 ± 0.16 d	74.19 ± 1.42 b	12.93 ± 0.64 c	7.26 ± 0.17 d
	6.23	-	-	14.44 ± 0.78 b	101.71 ± 1.05 a	14.22 ± 0.83 b	16.04 ± 1.16 b
	6.30	-	-	20.49 ± 0.14 c	118.64 ± 1.89 a	25.91 ± 1.16 b	29.07 ± 1.46 b
	7.7	-	-	44.21 ± 0.82 b	134.31 ± 1.14 a	32.55 ± 0.84 c	35.26 ± 0.91 c
	7.13	-	-	55.80 ± 0.76 a	-	35.14 ± 1.37 b	36.55 ± 0.69 b
	7.20	-	-	61.94 ± 1.02 a	-	35.45 ± 0.79 c	41.55 ± 0.95 b
	7.27	-	-	81.96 ± 0.53 a	-	44.60 ± 1.20 c	58.30 ± 1.04 b
	8.4	-	-	105.45 ± 2.67 a	-	65.22 ± 0.45 c	78.12 ± 0.71 b
Protein content (mg/g)	6.10	44.63 ± 0.82 a	41.54 ± 0.96 a	37.34 ± 0.85 a	43.39 ± 3.86 a	28.72 ± 0.59 b	26.72 ± 1.80 b
	6.16	37.76 ± 4.63 b	40.41 ± 1.93 b	39.51 ± 3.43 b	41.83 ± 4.22 b	48.46 ± 1.08 b	60.18 ± 0.96 a
	6.23	-	-	47.92 ± 0.44 b	88.83 ± 5.95 a	49.65 ± 0.63 b	46.13 ± 1.75 b
	6.30	-	-	43.86 ± 1.12 a	33.80 ± 4.69 b	40.18 ± 0.60 ab	48.67 ± 0.20 a
	7.7	-	-	43.36 ± 1.68 a	41.15 ± 5.65 a	46.31 ± 2.77 a	56.63 ± 4.88 a
	7.13	-	-	59.36 ± 0.70 c	-	81.42 ± 3.79 a	72.86 ± 1.64 b
	7.20	-	-	69.40 ± 1.63 b	-	83.31 ± 5.05 a	81.87 ± 0.85 b
	7.27	-	-	85.67 ± 1.20 a	-	72.75 ± 0.69 b	82.49 ± 1.89 a
	8.4	-	-	93.07 ± 1.16 a	-	77.40 ± 1.18 b	91.20 ± 0.73 a

Different letters next to the values (mean ± SE) in the same row indicate significant differences for amino acids content and protein content among different nectarine varieties on the same collection date that was examined using the S-N-K test (*p* < 0.05).

**Table 7 insects-10-00221-t007:** The soluble sugar and pectin content of nectarine varieties.

Index	Date	Shuguang	Chunguang	Fengguang	Zhongyou 4	Zhongyou 7	Zhongyou 8
Soluble sugar content (mg/g)	6.10	55.34 ± 1.05 a	51.51 ± 1.90 b	6.46 ± 0.73 d	23.17 ± 1.05 c	2.25 ± 0.25 e	2.55 ± 0.35 e
	6.16	62.21 ± 0.79 b	64.38 ± 1.20 a	15.54 ± 0.15 d	37.27 ± 0.69 c	2.83 ± 0.46 e	3.27 ± 0.31 e
	6.23	-	-	15.87 ± 0.54 b	38.65 ± 1.85 a	3.40 ± 0.14 c	3.37 ± 0.16 c
	6.30	-	-	16.91 ± 0.65 b	42.09 ± 0.71 a	17.28 ± 2.27 b	12.60 ± 0.70 b
	7.7	-	-	18.17 ± 1.00 b	51.44 ± 2.23 a	21.07 ± 0.62 b	16.79 ± 0.15 b
	7.13	-	-	26.09 ± 1.19 a	-	27.93 ± 1.45 a	20.96 ± 1.34 b
	7.20	-	-	27.80 ± 0.68 b	-	37.02 ± 1.44 a	21.87 ± 1.82 c
	7.27	-	-	36.69 ± 0.97 b	-	48.06 ± 1.02 a	24.61 ± 1.13 c
	8.4	-	-	45.05 ± 1.75 b	-	55.07 ± 2.15 a	30.55 ± 1.88 c
Pectin content (mg/g)	6.10	9.79 ± 0.97 b	12.75 ± 0.83 ab	6.33 ± 0.17 b	6.53 ± 1.41 ab	6.92 ± 0.77 a	4.72 ± 0.16 b
	6.16	14.41 ± 0.41 a	16.14 ± 0.44 a	8.65 ± 0.06 b	8.10 ± 1.55 a	8.06 ± 0.31 c	4.89 ± 0.90 d
	6.23	-	-	10.11 ± 0.62 a	11.47 ± 1.92 a	8.21 ± 0.59 a	7.30 ± 1.04 a
	6.30	-	-	10.91 ± 2.15 b	13.08 ± 0.54 c	8.82 ± 0.83 a	9.83 ± 0.47 a
	7.7	-	-	11.32 ± 2.21 ab	14.00 ± 1.18 b	9.21 ± 0.57 b	11.25 ± 2.00 a
	7.13	-	-	11.32 ± 2.32 a	-	10.19 ± 0.96 b	11.73 ± 2.00 b
	7.20	-	-	13.46 ± 0.95 a	-	14.11 ± 2.31 b	12.97 + 2.76 a
	7.27	-	-	14.17 ± 1.19 a	-	14.78 ± 1.18 a	13.39 ± 2.00 a
	8.4	-	-	14.26 ± 1.08 a	-	20.17 ± 0.45 a	16.74 ± 0.10 b

Different letters next to the values (mean±SE) in the same row indicate significant differences for soluble sugar content and pectin content among different nectarine varieties on the same collection date that was examined using the S-N-K test (*p* < 0.05).

**Table 8 insects-10-00221-t008:** Pearson correlation analyses between the amount of eggs and physiological indexes.

Index	The Amount of Eggs (Intact Nectarine)	The Amount of Eggs (Sliced Nectarine)
firmness	−0.83, <0.01	−0.90, <0.01
soluble solids content	1.00, 0.001	1.00, <0.002
amino acid content	0.57, <0.01	0.70, <0.01
protein content	0.30, 0.08	0.42, 0.01
soluble sugar content	0.74, <0.01	0.80, <0.01
pectin content	0.02, 0.89	−0.12, 0.48

The data indicate *Pearson* correlation coefficient and *p* value.

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
