# Peer review of "Oviposition Suitability of Drosophila Suzukii (Diptera: Drosophilidae) for Nectarine Varieties and Its Correlation with the Physiological Indexes"

_insects, 2019, doi:10.3390/insects10080221_

Round 1
Reviewer 1 Report
The study reports the occurrence of oviposition by Drosophila suzukii on several different nectarine cultivars in China under laboratory tests and shows that the oviposition preference is related to some external (e.g., firmness) or internal (e.g., soluble solid content sugar content, amino acid content, sugar content) fruit characteristics. To my knowledge, nectarine has not bene widely reported as a SWD host in its native or invaded regions. However, this study demonstrated that all ripe nectarine fruit are susceptible to the fly infestation, regardless the fruit varieties. Further, the authors suggest that mixture cultivation of different varieties with subsequent ripening periods could create local population movement on a micro-geographic scale and should be avoided. Overall, the manuscript is well written, the experiments were carefully conducted, and the data are appropriately analyzed. I suggest the acceptance of manuscript for publication with minor revision. Following is my specific comments for the authors to consider:
I like the introduction as it is straight forward and short. Today, the literature on SWD is accumulating rapidly, and covers almost every topic. My only suggestion is to focus on the discussions of various external (e.g., skin firmness, trichomes) or internal (e.g., sugar content, pH values) fruit characteristics that can affect the susceptibility of the fruit to SWD (e.g., Lee et al. 2011, Burrack et al. 2013, Steward et al. 2014). The authors start with very general discussion and in one place even mention parasitoid’s preference which is not relevant to this topic. A few of citations did not cite the original work or not the right citation. For example:
L33: feed (feeds).
L39: With regard to the economic loss, these two (5-6) are not the proper use of citations.
L40: “up to 80% economic loss for grape…? (10)”. SWD is still not considered a pest of grapes except a few susceptible varieties in some regions and this cited study is an risk assessment in Australia if the fly is invaded there.
L50: “Insect’s preference for oviposition substrate is closely related to color, variety and maturity of the host” (could fruit fly’s…host fruit).
L58: “of host plants affect the ovipositional preference of parasitoid”.
Materials and Methods: The results would be better if the authors directly count the number of eggs instead of larvae for the no choice test and measure the number of eggs successfully developed into adults. This will test if there is a positive link between adult’s oviposition preference and offspring performance.
L83: Please also indicates the year of fruit collections.
L92: How were the fruit picked up? Were they picked up uing a pruning shear to cut them off just above the stem-end; Because the area around the petiole is much soft and the petiole may naturally prevent the fly’s access to those more susceptible fruit surface? Also hand-picked fruit could remove small amount of the skin (see Steward et al. 2014). Please be more specific. This is also why I suggest counting eggs so that you know exact where the eggs are laid on the fruit surface.
L104: which part of the fruit surface was measured for firmness? Please be specific as firmness also vary with the location of surface.
Results: For all statistical results, please indicate the freedom of degree. I suggest providing an additional table to show all these correlation analyses results for easy grasp. You may also point out that the relationship between oviposition preference and fruit firmness is positive related not only across different varieties but also within the same varieties at different maturing stages (e.g. Fengguang).
Discussion: I disagree with the general argument of which variety being more susceptible than other ones. To me, all varieties are susceptible to the fly infestation based on your results after the fruit have ripen. It depends on the spatial and temporal dynamics of fly population in the region. For example, if the first generation comes out in early June, then these early mature varieties will be hit by the fly. The key is to avoid creating local population movement among differently-matured cultivars.
List references are not in the same format and citation 4 is incomplete (no journal title).
Author Response
Response to Reviewer 1 Comments
Point 1: L33: feed (feeds).
Response 1: Thanks for your suggestion, the word “feed” has been revised as “feeds”.
Point 2: L39: With regard to the economic loss, these two (5-6) are not the proper use of citations.
Response 2: Thanks for your suggestion, we have replaced the reference 6.
Point 3: L40: “up to 80% economic loss for grape…? (10)”. SWD is still not considered a pest of grapes except a few susceptible varieties in some regions and this cited study is an risk assessment in Australia if the fly is invaded there.
Response 3: Thanks for your suggestion, we have deleted this sentence.
Point 4: L50: “Insect’s preference for oviposition substrate is closely related to color, variety and maturity of the host” (could fruit fly’s…host fruit).
Response 4: Thanks for your suggestion, we have added some host fruits in the location.
Point 5: L58: “of host plants affect the ovipositional preference of parasitoid”.
Response 5: Thanks for your suggestion, we have changed the expression in the sentence.
Point 6: Materials and Methods: The results would be better if the authors directly count the number of eggs instead of larvae for the no choice test and measure the number of eggs successfully developed into adults. This will test if there is a positive link between adult’s oviposition preference and offspring performance.
Response 6: Thanks for your suggestion, due to the eggs of D. suzukii are white and transparent, they are difficult to count in sliced nectarine fruit, we count the number of larva only, we think the number of larva is more meaningful for this experiment.
Point 7: L83: Please also indicates the year of fruit collections.
Response 7: Thanks for your suggestion, we have added some information in the paragraph.
Point 8: L92: How were the fruit picked up? Were they picked up uing a pruning shear to cut them off just above the stem-end; Because the area around the petiole is much soft and the petiole may naturally prevent the fly’s access to those more susceptible fruit surface? Also hand-picked fruit could remove small amount of the skin (see Steward et al. 2014). Please be more specific. This is also why I suggest counting eggs so that you know exact where the eggs are laid on the fruit surface.
Response 8: Thanks for your suggestion, we have added some information about fruit picked up in the paragraph.
Point 9: L104: which part of the fruit surface was measured for firmness? Please be specific as firmness also vary with the location of surface.
Response 9: Thanks for your suggestion, the location for firmness is added in the paragraph.
Point 10: Results: For all statistical results, please indicate the freedom of degree. I suggest providing an additional table to show all these correlation analyses results for easy grasp. You may also point out that the relationship between oviposition preference and fruit firmness is positive related not only across different varieties but also within the same varieties at different maturing stages (e.g. Fengguang).
Response 10: Thanks for your suggestion, we have added some results information about correlation analyses in the Table 7.
Point 11: Discussion: I disagree with the general argument of which variety being more susceptible than other ones. To me, all varieties are susceptible to the fly infestation based on your results after the fruit have ripen. It depends on the spatial and temporal dynamics of fly population in the region. For example, if the first generation comes out in early June, then these early mature varieties will be hit by the fly. The key is to avoid creating local population movement among differently-matured cultivars.
Response 11: Thanks for your suggestion, we have changed some expression in the discussion section.
Point 12: List references are not in the same format and citation 4 is incomplete (no journal title).
Response 12: Thanks for your suggestion, we have rechecked carefully and changed the references format according to the journal.

Reviewer 2 Report
The manuscript is interesting and presents new information. The information of this manuscript can help in the selection of varieties of nectarines to be cultivated aiming the management of populations of SWD.
The work would have been even better if the authors had also carried out experiments with a chance to choose and evaluated the biology of D.suzukii in the different varieties.
I present below some corrections for text improvement.
Line 38 To replace D. suzukii is widely distributed all over the world by it is widely distributed in North America, Europe, and South America, and it is expected to establish itself in Africa and Oceania (Dos Santos et al. 2017)
Line 38-39 To replace (...) and causes serious loss to the fruit industry, such as damage to cherry crops in North America and Europe [5-6] by and causes serious loss to the fruit industry, such as damage to several crops in North America, Europe [5-6] and Brazil (Schlesener et al.,2019)
Schlesener, DCH,,Wollmann, J, Pazini, JB, Padilha, AC, Grützmacher, AD, Garcia,FRM. Insecticide Toxicity to Drosophila suzukii (Diptera: Drosophilidae) parasitoids: Trichopria anastrephae (Hymenoptera: Diapriidae) and Pachycrepoideus vindemmiae (Hymenoptera: Pteromalidae), Journal of Economic Entomology, , toz033, https://doi.org/10.1093/jee/toz033
Line 53 to 55 To delete Similarly, the Bactrocera tryoni had oviposition preference for fruit with softer pericarp [18], while B. oleae prefers to oviposit on olives with higher maturity [19].
Line 62 To delete Protein content can inhibit oviposition of Rhagoletis pomonella [26].
Line 346 To include the sentence (......)show higher resistance to D. suzukii than other varieties, similar results were found with Anastrepha fraterculus in peach in Brazil (Araujo et al.2019).
Reference
Araujo, E. S.; Paiva, L. R.; Alves, S. G.; Bevacqua, D.; Nava, D. E.; Lavigne, C.; Garcia, F. R. M. (2019). Phenological asynchrony between the fruit fly Anastrepha fraterculus and early maturing peach cultivars could contribute to pesticide use reduction. Spanish Journal of Agricultural Research, Volume 17, Issue 1, e1001. https://doi.org/10.5424/sjar/2019171-13294
Author Response
Response to Reviewer 2 Comments
Point 1: Line 38 To replace D. suzukii is widely distributed all over the world by it is widely distributed in North America, Europe, and South America, and it is expected to establish itself in Africa and Oceania (Dos Santos et al. 2017)
Response 1: Thanks for your suggestion, this sentence has been replaced.
Point 2: Line 38-39 To replace (...) and causes serious loss to the fruit industry, such as damage to cherry crops in North America and Europe [5-6] by and causes serious loss to the fruit industry, such as damage to several crops in North America, Europe [5-6] and Brazil (Schlesener et al., 2019)
Response 2: Thanks for your suggestion, we have replaced the sentence.
Point 3: Line 53 to 55 To delete Similarly, the Bactrocera tryoni had oviposition preference for fruit with softer pericarp [18], while B. oleae prefers to oviposit on olives with higher maturity [19].
Response 3: Thanks for your suggestion, we have deleted the two sentences and the references for [18] and [19] are also deleted.
Point 4: Line 62 To delete Protein content can inhibit oviposition of Rhagoletis pomonella [26].
Response 4: Thanks for your suggestion, we have deleted the sentence and the reference for [26] is also deleted.
Point 5: Line 346 To include the sentence (......) show higher resistance to D. suzukii than other varieties, similar results were found with Anastrepha fraterculus in peach in Brazil (Araujo et al.2019).
Response 5: Thanks for your suggestion, the sentence is added following the sentence “(......) show higher resistance to D. suzukii than other varieties”.
